# Genetic erosion reduces biomass temporal stability in wild fish populations

Jérôme G. Prunier [1] ✉, Mathieu Chevalier[2,5], Allan Raffard[1,6], Géraldine Loot[3], Nicolas Poulet[4] & Simon Blanchet [1,3] ✉

Genetic diversity sustains species adaptation. However, it may also support key ecosystems functions and services, for example biomass production, that can be altered by the worldwide loss of genetic diversity. Despite extensive experimental evidence, there have been few attempts to empirically test whether genetic diversity actually promotes biomass and biomass stability in wild populations. Here, using long-term demographic wild fish data from two large river basins in southwestern France, we demonstrate through causal modeling analyses that populations with high genetic diversity do not reach higher biomasses than populations with low genetic diversity. Nonetheless, populations with high genetic diversity have much more stable biomasses over recent decades than populations having suffered from genetic erosion, which has implications for the provision of ecosystem services and the risk of population extinction. Our results strengthen the importance of adopting prominent environmental policies to conserve this important biodiversity facet.

Biodiversity sustains critical ecosystem services, such as water filtering, pollination or biomass production[1], that are directly compromised by the ongoing global biodiversity crisis[2]. By promoting trait complementarity or redundancy among species, interspecific diversity allows ecological communities to optimally capture essential resources, to transform those resources into biomass and to recycle them[3,4]. In species-rich communities, these ecological processes are maintained even in the face of environmental fluctuations, thus promoting ecosystem productivity and stability over time:[5,6] this is the insurance effect of species richness[7].

Although biodiversity erosion is often associated to species loss, another form of erosion is silently underway: the loss of intraspecific genetic diversity[8]. Intraspecific genetic diversity can play a role similar to species diversity in driving ecological processes at the basis of ecosystem services, such as biomass production[9,10]. Beyond its positive influence on individual fitness and thus on per capita biomass production, intraspecific genetic diversity may favor functional complementarity or redundancy among individuals, thereby fostering a

more efficient exploitation of available resources[1,7,9]. Genetically diversified populations are therefore predicted to harbor both larger individuals (higher per capita biomass) and higher and more stable levels of total biomass than genetically impoverished populations[9,10]. This direct relationship between intraspecific genetic diversity and biomass is expected to be particularly strong in ecosystems where species diversity is naturally low, which is actually the norm in many temperate ecosystems[11]. In such cases, the functioning of ecosystems probably depends more on the complementarity among genotypes than on the complementarity among species[12,13], emphasizing the importance of maintaining genetic diversity to preserve ecosystem functions and services[9].

Most studies investigating the relationship between intraspecific genetic diversity and key-ecological parameters such as biomass are based on experimental or semi-experimental settings, where population densities and levels of intraspecific genetic diversity are manipulated, while environmental conditions are controlled and maintained constant over time[9,14]. However, observational studies conducted in

[1]Centre National de la Recherche Scientifique (CNRS), Université Paul Sabatier (UPS); Station d'Ecologie Théorique et Expérimentale, UAR 2029, F-09200 Moulis, France. [2]Department of Ecology and Evolution, University of Lausanne, Biophore, CH-1015 Lausanne, Switzerland. [3]CNRS, UPS, École Nationale de Formation Agronomique (ENFA), UMR 5174 EDB (Laboratoire Évolution & Diversité Biologique), 118 route de Narbonne, F-31062, Toulouse, cedex 4, France. [4]Pôle écohydraulique AFB-IMT, allée du Pr Camille Soula, 31400 Toulouse, France. [5]Present address: Ifremer, DYNECO, F-29280 Plouzané, France. [6]Present address: Univ. Savoie Mont Blanc, INRAE, CARRTEL, 74200 Thonon-les-Bains, France. ✉e-mail: jerome.prunier@gmail.com; simon.blanchet@sete.cnrs.fr

natural settings are still scarce and mostly concern plants[15]. Although these studies offer a number of advantages, experiments do not make it possible to cover large spatial and temporal scales or to investigate the influence of historical contingencies. Local levels of intraspecific genetic diversity indeed result from the interplay between long-term evolutionary trajectories (e.g., localization and size of glacial refugia[16]) and more recent –if not ongoing– ecological processes affecting individual life history traits or population demography (e.g., stressful environmental conditions[17], bottleneck events[18], or strong directional selection[19]). This natural complexity cannot be fully grasped by experimental studies. Observational field surveys are on the contrary more realistic and may provide important insights into the contribution of intraspecific genetic diversity, and the loss of it, to biomass and biomass stability in natural settings[15]. They yet raise several difficulties. First, assessing the influence of genetic diversity on biomass and biomass stability over several generations or seasonal cycles implies long-term monitoring programs of both population density and biomass, but such data are usually difficult to collect and are still scarce. Furthermore, the relationships between genetic diversity and biomass in across-population studies may be masked by the interplay with other factors also involved in biomass production, such as population density and environmental conditions, making it difficult to disentangle their respective contributions[15]. This last issue may, however, be partly alleviated through the use of causal modeling procedures (path analyses), making it possible to thoroughly confront theoretical expectations and experimental findings with the real world[3,20].

Here, we capitalized on long-term field surveys of three parapatric non-commercial freshwater fish species (*Phoxinus dragarum*, *Gobio occitaniae* and *Squalius cephalus*) from two large river basins in southwestern France (Fig. 1) to assess the relationships between intraspecific genetic diversity, total fish biomass and biomass temporal stability (measured as the inverse of biomass variability[21]), while controlling for the effect of environment (upstream-downstream gradient and eutrophication levels), of per capita biomass (or its temporal stability) and of past demographic events, using path analyses[22]. Total fish biomass stood for the total weight of all individuals from the three focal species collected at a given site (in g.m$^{-2}$), standardized and then averaged across species and over years. Local levels of intraspecific genetic diversity were computed for each species using both microsatellite and SNPs data and similarly averaged across species. We addressed the following questions: Do the positive relationships found experimentally between intraspecific genetic diversity and total biomass and total biomass stability hold true in natural settings? If any, is the contribution of intraspecific genetic diversity to these ecosystem functions comparable in magnitude to that of other environmental determinants? Finally, is it possible to detect the impact of contemporary genetic erosion -i.e., the loss of intraspecific genetic diversity in response to a recent reduction in population size- on biomass and biomass stability of fish populations? This latter point is of high concern: with conservative estimates of 6-15% loss of intraspecific genetic diversity in wild organisms in the Anthropocene[23,24], the impact of human-induced genetic erosion on natural ecosystems' capacity to provide critical provisioning and regulating services to humanity may actually be much more important than anticipated. We show that populations with high genetic diversity do not reach higher biomasses than populations with low genetic diversity, but that they have much more stable biomasses over recent decades than populations having suffered from genetic erosion.

## Results

### Drivers of intraspecific genetic diversity
We found that intraspecific genetic diversity increased downstreamward (Figs. 2 and 3b; Table 1), a classical pattern in rivers that could stem from asymmetrical gene flow, the presence of glacial refugees and/or higher effective population sizes in downstream

areas[25]. Intraspecific genetic diversity was also indirectly impacted by water eutrophication, through a higher probability of having suffered from a bottleneck as eutrophication increases (Figs. 2 and 3a). As expected, the bottleneck probability altered spatial patterns of intraspecific genetic diversity: the loss of intraspecific genetic diversity associated with recent bottlenecks was particularly strong in downstream populations (Fig. 3b). This context-dependency of contemporary genetic erosion may reflect the observation that downstream areas are usually subject to multiple stressors (pollution, urbanization, channelization, …) that may reinforce the link between recent demographic changes and intraspecific genetic diversity.

### Drivers of total biomass
We found no significant relationship between intraspecific genetic diversity and either total biomass or per capita biomass (Fig. 2a; Table 1). Yet, total biomass was directly linked to per capita biomass (Fig. 3d), indicating that total biomass stemmed from the presence of (a few) large individuals rather than that of many small individuals. Per capita biomass decreased downstreamward and increased with eutrophication (Fig. 3c), suggesting that, although eutrophication may indirectly have a long-term negative influence on populations and associated intraspecific genetic diversity, it may locally boost the overall system productivity by favoring individuals' body condition[26].

### Drivers of biomass stability
Contrastingly, we found that fish populations with higher levels of intraspecific genetic diversity displayed higher biomass stability over time than genetically-impoverished populations, whatever the temporal stability in per capita biomass (i.e., positive relationship between biomass stability and intraspecific genetic diversity: Figs. 2b and 3f; Table 1). This important finding also held true when each species was analyzed separately (Supplementary Table 1). By favoring functional complementarity or redundancy among phenotypes[9,10,27], higher genetic diversity likely allows populations to maintain an efficient exploitation of available resources in the face of natural environmental fluctuations, ensuring a stable production of biomass[1,7]. Biomass stability also tended to decrease with eutrophication (Figs. 2b and 3f; Table 1), probably because biomass stability was negatively related to total biomass (Supplementary Figure 1), and thus indirectly to per capita biomass, the latter increasing with eutrophication (Figs. 2a and 3). Fish biomasses were thus higher in the most eutrophic sites, but they were less stable over time: were this finding to be confirmed by further studies, this dual and opposite effect of eutrophication would have important implications for the conservation of fish populations, since population stability is generally associated with lower extinction risk[28]. It is also noteworthy that temporal fluctuations in total fish biomass were unrelated to fluctuations in per capita biomass (that is, to temporal fluctuations in the average mass of individuals), reinforcing the hypothesis that biomass stability was rather indirectly fostered by mechanisms such as functional complementarity or redundancy among phenotypes[9,10,27], acting as a biological insurance against natural environmental fluctuations[7].

### Contribution of intraspecific genetic diversity to biomass stability
The contribution of intraspecific genetic diversity to the overall variance in biomass stability ($R^2$ = 21.1%) was much higher than that of considered environmental determinants ($R^2$ = 4%; Fig. 4). 84 % of the total explained variance in fish biomass stability was hence attributed to intraspecific genetic diversity, which suggests that intraspecific genetic diversity is a substantial driver of biomass stability in this area, although other unmeasured variables also likely sustain variation in biomass stability since a non-negligible part of this variation (-75%; Fig. 4) remained unexplained by our model. Moreover, first-order

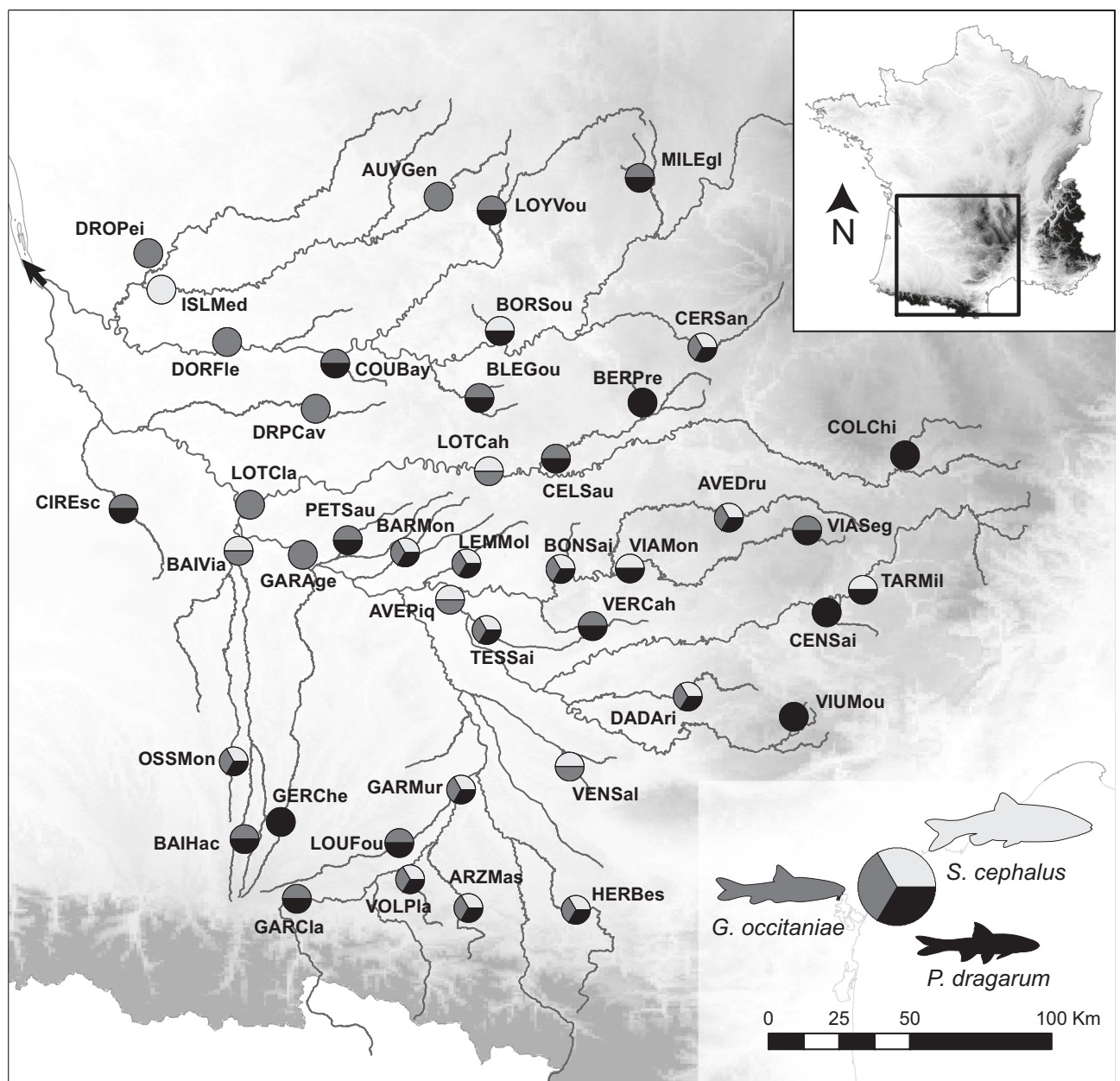

**Fig. 1 | Study area and localization of sampled river stations and species.** Geographic situation of the Garonne-Dordogne River basin in southwestern France and localization of the 42 unique river stations, with pie charts indicating species (co-)occurrence within each station. The black arrow indicates the location of the river mouth. Background is a shaded relief map. Source data are provided as a Source Data file.

interactions between intraspecific genetic diversity and environmental variables were not retained in final models (Fig. 2; but see the specific case of minnows in Supplementary Table 1c and Supplementary Figure 2), suggesting that the influence of intraspecific genetic diversity on total biomass stability may be predictable across environmental gradients, a result which is yet to be generalized to different taxa and ecosystems.

## Discussion

Capitalizing on long-term demographic surveys, we report a positive relationship between intraspecific genetic diversity and temporal biomass stability in three freshwater fish species. This relationship indicates a buffering effect of intraspecific genetic diversity, genetically-impoverished populations being less efficient in maintaining stable biomass levels over time than genetically-diversified populations[7]. By favoring higher functional complementarity among

phenotypes, higher genetic diversity likely allows populations to maintain an efficient exploitation of available resources in the face of natural environmental fluctuations, ensuring a stable production of biomass[1,4,7]. Interestingly, this buffering effect of intraspecific genetic diversity did not come with a performance-enhancing effect on biomass production[7] (Fig. 2a): genetically-diversified populations did not show higher biomass levels compared to genetically-impoverished populations, mean total biomass being mostly driven by per capita biomass, and indirectly by the environment (Fig. 4). Nevertheless, our study provides one of the first non-experimental evidence that real-world genetic diversity can directly promote temporal stability in biomass of wild organisms, in line with both theoretical expectations and experimental evidence[9,14].

Our study being based on empirical data, it is not surprising that a large amount of variance in mean biomass and in biomass stability remained unexplained by our models (60 and 75%,

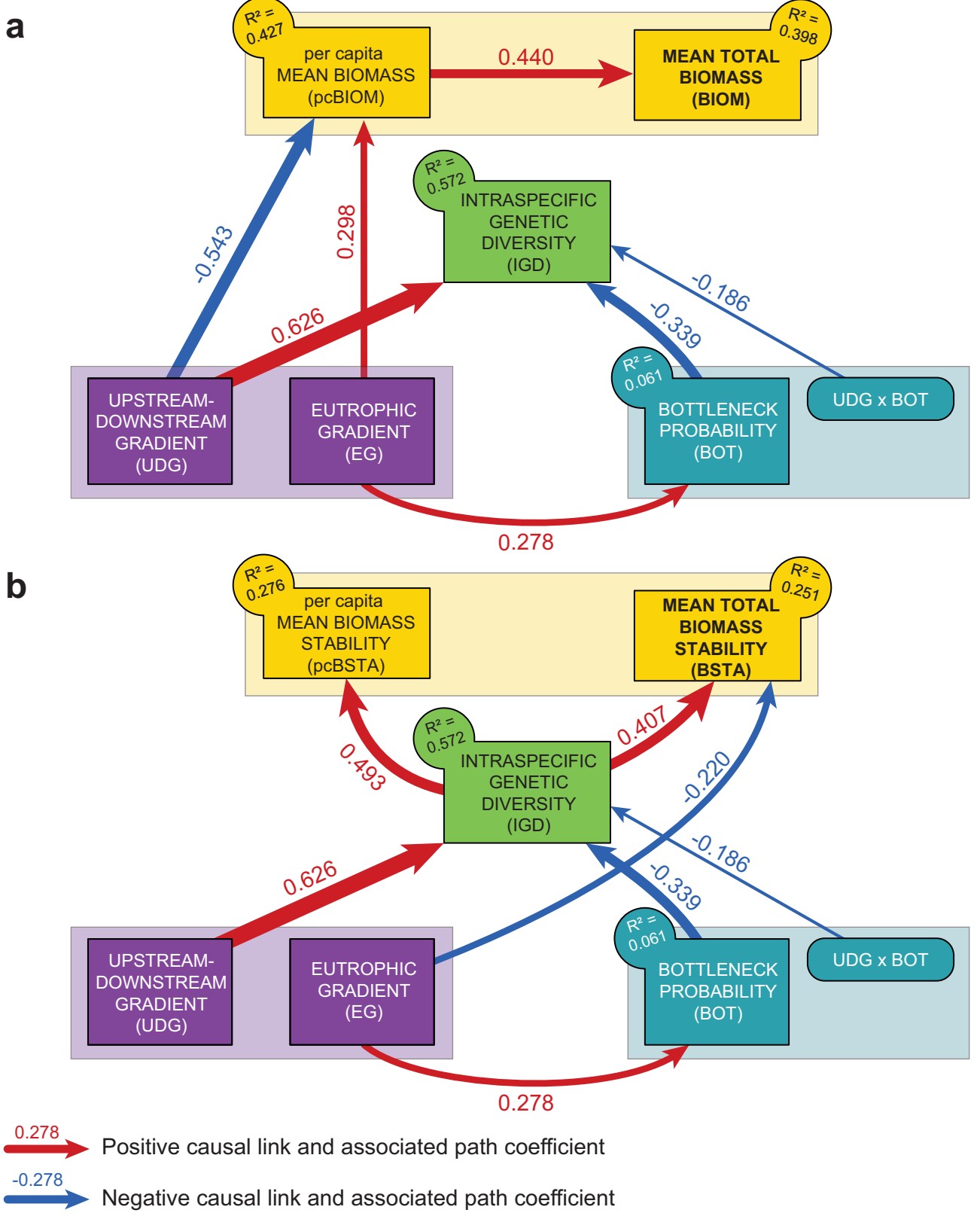

**Fig. 2 | Final causal graphs. a** Causal graph depicting the retained links among environmental (purple), bottleneck (blue), genetic (light green), and mean biomass variables (per capita and total; yellow). **b** Causal graph depicting the retained links among environmental (purple), bottleneck (blue), genetic (light green), and biomass stability variables (per capita and total; yellow). In each panel, retained first-order interactions are represented by rounded rectangles. Blue and red arrows represent negative and positive significant paths, respectively, with the width of arrows proportional to the absolute value of the corresponding path coefficient. Also provided is the amount of variance (R²) explained in each endogenous variable. Non-significant paths are not represented, for the sake of clarity (see Table 1 for detailed results). Source data are provided in Table 1.

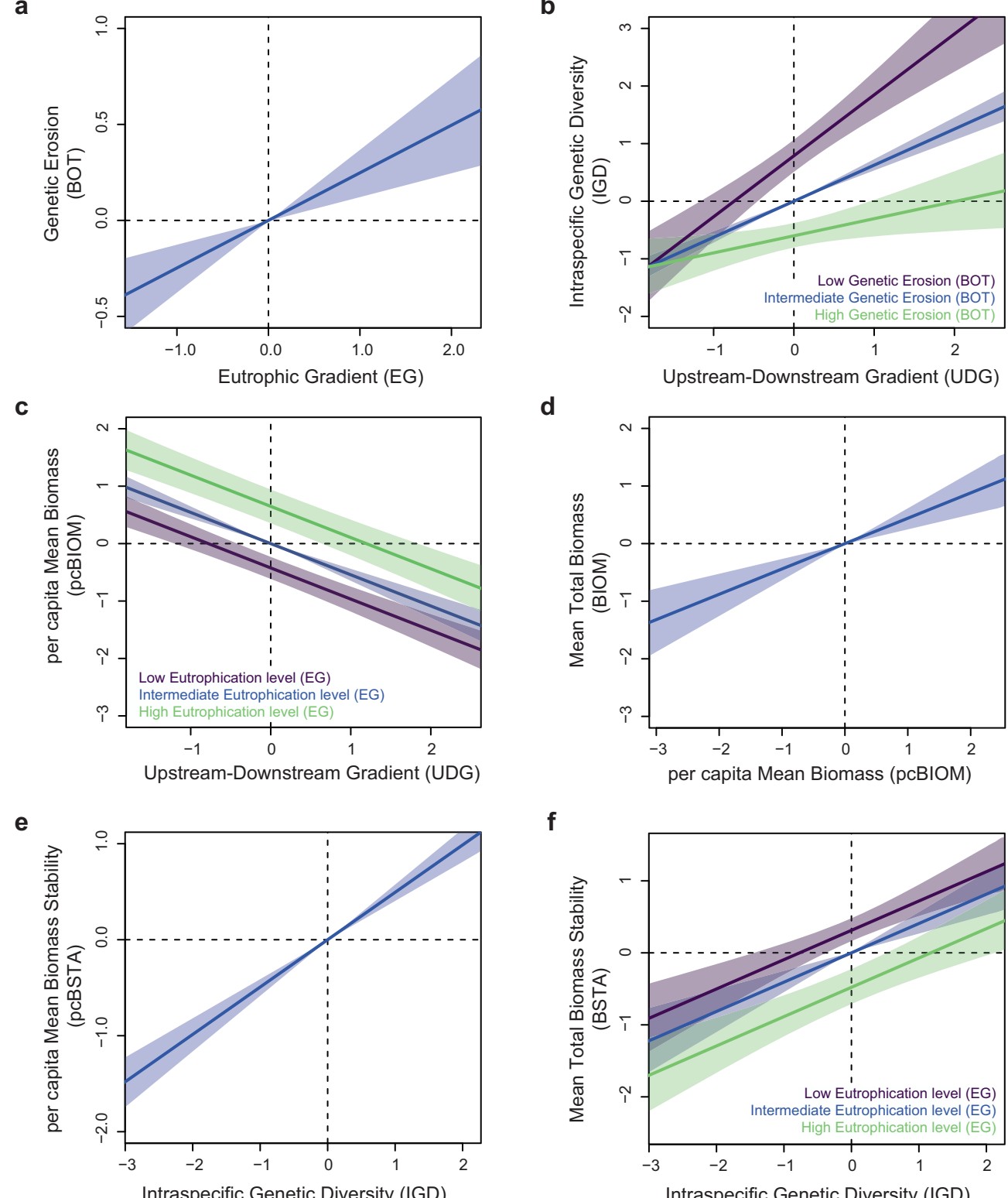

**Fig. 3 | Predicted values of all endogenous variables given the retained links shown in Fig. 1.** Data are presented as predicted values (thick lines) +/- SD (colored envelops). **a**: predicted values for Genetic Erosion (bottleneck probability BOT); **b**: predicted values for Intraspecific Genetic diversity (IGD); **c**: predicted values for per capita Mean Biomass (pcBIOM); **d**: predicted values for mean Total Biomass (BIOM); **e**: predicted values for per capita Mean Biomass Stability (pcBSTA); **f**: predicted values for Mean Total Biomass Stability (BSTA). When two predictors were considered (panels **b, c** and **f**), predictions were performed using the full range of the first predictor (x-axis) and the minimal (purple), mean (blue) and maximal (green) values of the second predictor. Source data are provided as a Source Data file.

**Table 1 | Detailed results from simplified causal models**

| Predictor→ Response | Model A | Model B |
|---|---|---|
| | $\chi^2_{(14,42)}$ = 13.71, *p* value = 0.471 | $\chi^2_{(14,42)}$ = 9.22, *p* value = 0.817 |
| | CFI = 1 | CFI = 1 |
| | SRMR = 0.072 | SRMR = 0.075 |
| UDG→ BIOM | −0.275 [−0.585; 0.034] / {−1.74 | 0.081} | |
| pcBIOM→ BIOM | 0.440 [0.083; 0.797] / {2.415 | 0.016} | |
| R² | 0.398 | |
| **IGD→ BSTA** | | **0.407 [0.119; 0.695] / {2.772 | 0.006}** |
| EG→ BSTA | | −0.220 [−0.439; −0.001] / {−1.969 | 0.049} |
| R² | | 0.251 |
| UDG→ pcBIOM | −0.543 [−0.749; −0.338] / {−5.178 | 0} | |
| EG→ pcBIOM | 0.298 [0.032; 0.563] / {2.20 | 0.028} | |
| R² | 0.427 | |
| IGD→ pcBSTA | | 0.493 [0.325; 0.662] / {5.746 | 0} |
| EG→ pcBSTA | | −0.204 [−0.489; 0.081] / {−1.146 | 0.161} |
| R² | | 0.276 |
| UDG→ IGD | 0.626 [0.433; 0.820] / {6.341 | 0} | |
| BOT→ IGD | −0.339 [−0.576; −0.102] / {−2.804 | 0.005} | |
| UDG x BOT→ IGD | −0.186 [−0.346; −0.026] / {−2.276 | 0.023} | |
| R² | 0.572 | |
| EG→ BOT | 0.278 [0.006; 0.490] / {2.007 | 0.045} | |
| R² | 0.061 | |

For each model A (Mean Biomass) and B (Biomass Stability), the table provides absolute fit indices (CFI, SRMR, χ² statistic and associated two-tailed *p* value), amounts of explained variance (R²) in response variables and estimates of path coefficients with 95% confidence intervals obtained from bootstrap resampling (between square brackets) and associated statistics (z-values | *p* values, between curly brackets). The significant positive relationship between intraspecific genetic diversity IGD and Biomass stability is highlighted in bold. UDG: Upstream-downstream gradient; EG: Eutrophication gradient; IGD: Intraspecific genetic diversity; BOT: Bottleneck probability; BIOM: Total Biomass; BSTA: Total Biomass Stability; pcBIOM: per capita Mean Biomass; pcBSTA: per capita Mean Biomass Stability.

respectively; Figs. 2 and 4). Mean biomass and biomass stability were also probably driven by factors that we could not consider in this study, such as interspecific interactions at the community-level[10] or, at the ecosystem level, autotroph primary production[29] or terrestrial subsidies[30]. Nevertheless, intraspecific genetic diversity accounted for more than 21% of the variance in biomass stability, a contribution much higher than that of considered environmental predictors. Our findings not only indicate that the relationship between intraspecific genetic diversity and biomass stability holds true in natural ecosystems, but also that this relationship can be substantial compared to the effects of other undisputable determinants of biomass, as recently shown for interspecific diversity[5]. While species richness can buffer natural fish biomass production against environmental variations[20], we argue that the intraspecific facet of biodiversity may actually also contribute to biomass stability in the wild[9,14].

A corollary to this finding is that the loss of intraspecific genetic diversity might undermine the temporal stability in biomass production. We notably detected a significant negative relationship between intraspecific genetic diversity and the probability that populations experienced a recent demographic contraction (i.e., recent bottleneck, Fig. 2), which is consistent with the imprint of recent human activities on contemporary levels of genetic diversity[23,24]. This finding must be considered with caution, since recent bottleneck probabilities were estimated using a modest number of microsatellite markers, which can generate bias in demographic inferences[31–34]. Specifically, the limited number of loci sampled in the genome[31] and the departure from the assumed mutation model[32,33] or from mutation-drift equilibrium[34] generally increase the probability of inferring false signals of bottlenecks, i.e., of detecting a population decline in a truly stable population. This type of bias could have inflated the relationship we found between bottleneck probability and patterns of intraspecific genetic diversity. However, Paz-Vinas et al.[35]. demonstrated using simulations

that, in river systems, demographic inferences based on microsatellite markers are rather robust to this bias, and actually more likely to detect false signals of expansion, i.e., detect a population expansion in a truly stable population, than to detect false signal of bottlenecks. Furthermore, to somehow limit the potential biases associated with the use of microsatellite markers, past bottleneck inferences were based on three independent methods that yielded congruent estimates (Supplementary Table 2 and Supplementary Figure 3). Although the links between past-demographic events and contemporary patterns of intraspecific genetic diversity would merit further confirmation based on genomic data[31], we believe that it is reasonable to consider the low levels of intraspecific genetic diversity observed in some populations to stem -at least in part- from recent, possibly human-induced, demographic contractions, for instance triggered by water eutrophication (Figs. 2 and 3a). Although alternative and non-exclusive historical processes (e.g., postglacial colonization events) are also likely to explain observed spatial patterns of intraspecific genetic diversity[25,36], this suggests that contemporary evolutionary processes can modulate ecological dynamics in natural settings. Since the loss of intraspecific genetic diversity always precede the loss of species[8], genetic erosion may adversely affect key-ecological functions long before the first species of a community becomes extirpated. We therefore argue that the loss of intraspecific diversity observed worldwide[23,24] may actually be responsible for a considerable alteration of many ecological processes in nature, but that these adverse effects might have been underestimated. With a loss of 6-15% in intraspecific genetic diversity[23,24], we estimated a similar 8-10% reduction in biomass stability across the river basin ($\bar{D}_k$ = −8.9%; 95% confidence interval: [−10.1; −7.7]; Supplementary Figure 4). This reduction in biomass stability was calculated using conservative estimates of intraspecific genetic diversity loss[23,24]; we therefore anticipate that this reduction could be much greater in species with weak conservation statuses, and

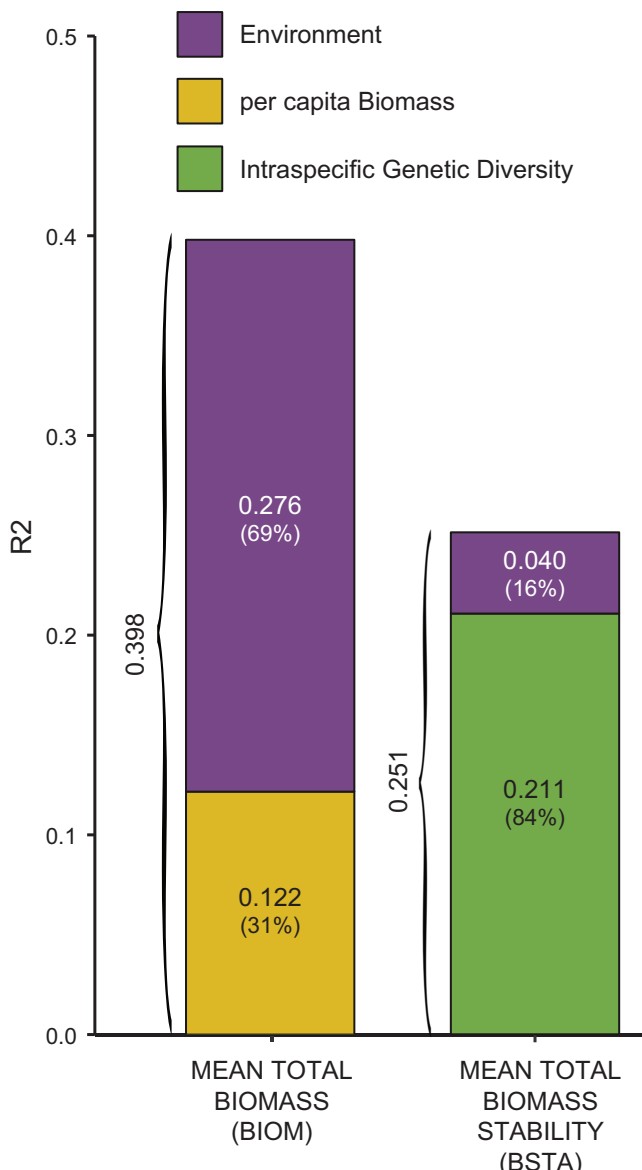

**Fig. 4 | Variance partitioning.** Contributions of environment variables (purple), per capita biomass variables (yellow) and intraspecific genetic diversity (light green) to the variance ($R^2$) in Total Biomass ($R^2 = 0.398$) and Total Biomass stability ($R^2 = 0.251$). The contributions to the explained variance (in %) are indicated into brackets. Source data are provided in the figure.

that this reduction in stability may have important cascading effects on trophic networks, ecosystem functioning and, in some countries, food provisioning[1,20].

Future studies are needed to confirm the significance of these results in other taxa and other ecosystems and to disentangle the relative contribution of intra- and interspecific diversity in explaining biomass production in the wild[10]. Nevertheless, our work suggests that the impact of genetic erosion on natural ecosystems' capacity to provide critical provisioning and regulating services to humanity is probably more important than anticipated. This makes human-induced genetic erosion a critical conservation issue and stresses the need for human societies to adopt prominent environmental policies favoring all facets of biodiversity[37].

## Methods

This study complies with all relevant ethical and permitting regulations. The field sampling protocol was approved by all the prefectures of the departments in which the sampling was carried out. The data generated in this study have been deposited in the Figshare database[38] under accession code https://doi.org/10.6084/m9.figshare.13095380.v9.

### Sampling stations and biological models

We selected 47 river stations evenly scattered across two large River basins in southwestern France (the Garonne River basin and the Dordogne River basin) to reflect the environmental variability existing along the upstream–downstream gradients. Fish communities in these basins are generally poorly diverse (3 to 15 species[39]), and we focused on the three most common species: the minnow *Phoxinus dragarum*, the gudgeon *Gobio occitaniae* and the chub *Squalius cephalus*. These generalist cyprinids vary in their mean body length (minnows: 80–90 mm; gudgeons: 120–150 mm; chubs: 300–500 mm)[40]. They mainly feed on invertebrates (although chubs can also consume small-bodied fish) but occupy different habitats: chubs are primarily found in downstream sections at relatively low densities (-0.01 ind.m$^{-2}$), minnows are primarily found in upstream sections at relatively high densities (-0.10 ind.m$^{-2}$), whereas gudgeons are found all along the river basin in various habitats and at relatively high densities (-0.08 ind.m$^{-2}$)[40]. All stations are monitored yearly by the French Office for Biodiversity (OFB) with a constant sampling effort since 1990[41]. Demographic and biomass data from the three focal species were extracted from the ASPE database[42]. We only retained stations monitored from 1993 to 2020 with at least eight sampling sessions, resulting in 42 stations (Fig. 1; mean number of sampling sessions = 15.6; mean survey duration = 21.2 years; mean number of focal species per station = 2.0). The minimum pairwise distance among sampling stations was 18.5 km (between TARMil and CENSai). This distance is higher than the maximum travelled distance recorded in chubs (16 km)[43], here considered as the most mobile species, and all stations could thus be considered independent.

### Biomass data

For each species, station and year of survey, we collected local fish density (number of individuals per m²), total fish biomass (in g.m$^{-2}$) and computed per capita biomass (or mean individual biomass, in g) as total fish biomass divided by local fish density. For each species, both total fish biomass and per capita biomass were standardized to make data comparable across species. For each station, we computed (i) Mean Total Biomass (respectively, per capita Mean Biomass) as the mean of total fish biomass (respectively, of per capita biomass) across species and over years, and to capture temporal fluctuations in biomass measures, (ii) Mean Total Biomass Stability (respectively, per capita Mean Biomass Stability) as the inverse of the squared coefficient of variation of Mean Total Biomass (respectively, of per capita Mean Biomass Stability) over years[44]. Per capita Biomass (stability) was here considered to determine whether Total Biomass (stability) was directly driven by the average individual biomass (stability) alone, or by other mechanisms such as functional complementarity among phenotypes. Each station was also assigned a sampling weight (ranging from 0.29 to 0.98) computed as the average of the relative local survey duration (compared to the maximal duration across stations) and of the relative local number of sampling sessions (compared to maximal number of sampling sessions across stations).

### Genetic sampling and extraction

The 42 retained stations were sampled in 2011 and 2014 with up to 30 adults from each species caught by electric-fishing, resulting in a set of 35, 37 and 21 sampled populations in *P. dragarum*, *G. occitaniae* and *S. cephalus*, respectively. On the field, a small piece of pelvic fin was collected from each individual and was preserved in 70% ethanol, before releasing fish in situ. For each individual, genomic DNA was extracted using a salt-extraction protocol[45].

## Microsatellite data

Genetic material collected in 2011 was used to genotype individuals at 18, 15 and 19 microsatellites markers in *P. dragarum*, *G. occitaniae* and *S. cephalus*, respectively. Polymerase chain reactions and genotyping were performed as detailed in Supplementary Data 1, resulting in a final dataset of 3262 genotypes (1177 in *P. dragarum*, 1227 in *G. occitaniae* and 858 in *S. cephalus*). We checked for multi-locus deviation from Hardy-Weinberg Equilibrium and for gametic disequilibrium using GENEPOP 4.2.1[46] after sequential Bonferroni correction to account for multiple related tests[47]. In each species, the presence of null alleles was assessed by analyzing homozygote excess at each locus in five populations previously identified as panmictic, using MICROCHECKER 2.2.3[48]. We discarded from further analyses any locus showing significant gametic disequilibrium and/or evidence of null alleles, resulting in the withdrawal of one locus (CtoG-075) in *P. dragarum*, two loci (Lsou5 and Gob12) in *G. occitaniae* and three loci (Ca1, Lid11 and LleC-090) in *S. cephalus*, for a total number of 17, 13 and 16 loci in each species, respectively. Although the three focal species are of limited interest for anglers[49], discriminant analyses of principal components[50] performed on microsatellite data allowed identifying outlier populations, possibly resulting from past stocking events[51]. All outlier populations (one in *P. dragarum*, four in *G. occitaniae* and one in *S. cephalus*; Supplementary Figure 5) were subsequently discarded from further analyses. For each species, a single outlier population was yet considered for de novo genome assembly.

## SNP data

For each species and each station, DNA from all individuals was pooled at equimolar concentrations to reach a total amount of 5 mg of DNA. Individual concentrations were determined using a QuBit 2.0 fluorometer (2.0, Life Technologies, Carlsbad, CA, USA). In *P. dragarum* and *G. occitaniae*, pooled DNA from each population was homogenized and split into two replicates, for subsequent analysis of allelic frequencies reliability. Pooled DNA was digested using SbfI restriction enzymes, followed by barcode ligation, sample pooling, DNA shearing, size selection of RAD tags (150 bp), adaptor ligation, RAD tag amplification and sequencing on two Hiseq lanes (GeT Platform, Toulouse, France). The procedure resulted in demultiplexed paired-end short reads that were subsequently processed for SNP identification and allelic frequencies estimation. All paired raw fastq files were filtered using the process_radtags and the clone_filter functions from Stacks[52], in order to remove reads with uncalled bases or low quality scores and discard PCR duplicates. For each species, a single fastq file, corresponding to an outlier population as identified from microsatellites data (AUVGen, TARMil and DRPCav in *P. dragarum*, *G. occitaniae* and *S. cephalus*, respectively; Supplementary Figure 5), was then processed using the Velvet de novo sequence assembler[53] to design a draft reference genome. Velvet's assembly parameters were optimized using the VelvetOptimiser wrapper[53], with 19 and 99 as starting and ending hash values, a minimum contig length of 150 pb and an insert length of 240pb. Draft genomes (in *fasta* format) were then indexed using both the *index* function from bwa[54] and the faidx function from SamTools[55]. All filtered paired-end *fastq* files were aligned on their draft genome using the aln and sampe functions from bwa. Aligned SAM files were converted to BAM format with the view and sort functions from SamTools, and filtered for unpaired, unmapped or badly mapped reads (mapping quality score <20) using the filter function from BamTools[56]. For each species, all indexed and filtered BAM files were then assembled in a single mpileup file using the mpileup function from SamTools. These mpileup files were synchronized in Popoolation2[57] with the mpileup2sync.jar java script. SNP allelic frequencies were finally determined using the snp-frequency-diff.pl perl script in Popoolation2 with a minimum allele count of 4 and a coverage ranging from 30 to 400. The whole procedure led to the identification of 10137, 13671 and 5897 SNPs in *P. dragarum*, *G.*

*occitaniae* and *S. cephalus*, respectively. In *P. dragarum* and *G. occitaniae*, allelic frequency reliability was assessed for each SNP and each station by comparing allelic frequencies between pairwise replicates. When allelic frequencies were available for the two replicates and when $\Delta_{AF}$, the difference in allelic frequencies between pairwise replicates, was lower than 0.25, the final allelic frequency was computed as the average of allelic frequencies across replicates. Otherwise, the final allelic frequency was set as missing data. In each species, we finally followed a two-step filtering procedure: (i) we first discarded any SNP with available allelic frequencies for less than 15 stations; (ii) we then discarded any station with available allelic frequencies for less than 150 SNPs. This final filtering procedure generated a total of 1244, 1892 and 1847 SNPs in 27, 30 and 17 populations in *P. dragarum*, *G. occitaniae* and *S. cephalus*, respectively (Supplementary Table 3).

## Metric of Intraspecific Genetic Diversity IGD

For each species and station, we computed a total of four metrics of genetic diversity. First, we used SNPs allelic frequencies to compute two metrics in R:[58] the expected level of heterozygosity across SNPs loci (sHe) and the observed level of SNP polymorphism (sPo), computed as the number of non-fixed loci (0 <allelic frequency <1) divided by the total number of loci with non-missing data in a given population. We then used microsatellite data to compute two additional metrics, the expected (μHe) and observed (μHo) levels of heterozygosity across microsatellite loci, using the software GENETIX 4.3[59]. These four metrics of genetic diversity naturally range between 0 and 1 and are thus directly comparable across species: for each station, we thus averaged each metric over species and then used a principal component analysis (PCA; R-package FactoMineR[60]) to get a synthetic predictor of the overall level of genetic diversity at the station level. Only the first principal component (PC) was retained, accounting for 75.9 % of variance in genetic data, with genetically impoverished sites on the one hand (negative coordinates) and genetically diversified sites on the other hand (Supplementary Figure 6).

## Bottleneck probability

We used microsatellite data to compute three different quantitative measures of the degree of genetic erosion that populations underwent in recent generations: the M-ratio[61], the N-ratio computed with Migraine[62] (hereafter, MIratio) and the N-ratio computed with VarEff[63] (hereafter, VEratio). Note that the M-ratio can only detect signals of population decline (bottlenecks), but that both the MIratio and the VEratio can also detect signals of population expansion.

The M-ratio is the ratio between the number of observed alleles at a microsatellite locus and the allelic range of that locus, the latter being supposed to decrease slower than the number of alleles during a demographic collapse. This index, ranging from 0 to 1, is inversely proportional to the degree of genetic erosion[61], and has been shown to be particularly relevant in river systems[35]. For each station and each species, the M-ratio was computed for each microsatellite locus and then averaged over loci.

The MIratio was computed as $\theta_{cur}/\theta_{anc}$, that is the ratio between the scaled current population size $\theta_{cur}$ and the scaled ancestral population size $\theta_{anc}$ as inferred with Migraine[62]. The MIratio was estimated using the OnePopVarSize model, considering a single past change in population size, and a generalized stepwise mutation model (GSM). For each station and each species, PAC-likelihood computations were based on four iterations, 500 points and iteratively 2000 (first computation) or 20,000 runs per point (second computation), to check consistency and improve convergence. In each case, we kept the estimate associated with the lowest RMS residual error, except when the algorithm failed to converge with the first computation, in which case we kept the second estimate.

The VEratio was similarly computed as $Ne_{cur}/Ne_{anc}$, that is the ratio between the estimated current effective population size $Ne_{cur}$

and the estimated ancestral effective population size $Ne_{anc}$ as inferred with the R-package VarEff[63]. For each station and each species, and following authors' recommendations, we first ran preliminary tests with short Markov chain Monte Carlo (MCMC) batches (1000 batches of length 1) to identify the best values for the number of past changes in effective population size (Jmax), for the effective size prior value (Nbar) and for the number of generations since the assumed origin of the population (Gbar). We then use these best values to run long MCMC batches (10,000 batches of length 10), get effective population sizes at generation 1 ($Ne_{cur}$) and Gbar ($Ne_{anc}$), and compute VEratio. All runs were performed with 10 spaces between batches, a burnin period of 10,000, a two-phase mutation model and a mutation rate of 0.0005.

MIratio and VEratio were log-transformed to meet normality assumptions. For each station, we averaged each metric over species and then used PCA to get a synthetic predictor (bottleneck probability BOT) of the overall level of genetic erosion at the station level, with negative coordinates corresponding to stations with low genetic erosion (high M-ratio, MIratio and VEratio values). Only the PC was retained, accounting for 61.3 % of variance (Supplementary Figure 3).

## Environment data

For each station, we computed the distance to the mouth (in m) and the distance to the tributary source (in m) with the riverdist R-package[64]. We also collected ten additional variables related to water quality, measured in June, July and August from 2000 to 2015 using the Water Information System of the Adour Garonne basin (SIE database): Temperature (in °C), oxygen concentration (in mg.L$^{-1}$) and saturation (in %), Biochemical oxygen demand (in mg.L$^{-1}$), as well as concentrations (in mg.L$^{-1}$) in nitrogen compounds (ammonium $NH_4^+$, nitrates $NO_3^-$ and nitrites $NO_2^-$), in phosphorus compounds (total phosphorus P and phosphate $PO_4^{3-}$) and in dissolved organic carbon. Following criteria used by French managers to assess the ecological status of rivers from various physicochemical parameters according to the French implementation of the European Water Framework Directive 2000/60/EC (Supplementary Figure 7), we assigned to each station, each month of survey and each water quality variable but temperature a value ranging from 1 (very good water quality) to 6 (very bad water quality). For each station, values were then averaged over water quality variables, then over months and finally over years to get a final Water Quality Index (WQI). The coefficients of variation of WQI over time did not exceed 0.43, indicating that water quality remained relatively stable over the considered period. The WQI theoretically ranges from 1 to 6, but here it ranged from 1 to 2.52 only (mean 1.35), indicating that all stations showed good to very good water quality. We used PCA to get synthetic predictors of environment characteristics (distance to the river mouth, distance to the tributary source, water temperature and Water Quality Index) at the site level (Supplementary Figure 8). The two first components were retained, accounting for 80.5 % of the total variance in environmental variables. The first component (55 % of variance) stood for the upstream-downstream gradient, with fresh upstream sites on the one hand (negative coordinates) and warmer downstream sites on the other hand. The second component (25.5 % of variance) stood for a eutrophic gradient, with oligotrophic river sites on the one hand (negative coordinates, low Water Quality Index) and nutrient-rich (mesotrophic) sites on the other hand (high Water quality Index).

## Path analyses

To investigate how intraspecific genetic diversity might influence Biomass and/or Biomass Stability while accounting for the effects of both environment and per capita Biomass[20,65], we used path analyses[66,67]. We designed two full causal models describing the expected direct and indirect links among variables and their first-order interactions (Supplementary Figure 9). Our main focus was on the direct links between intraspecific genetic diversity (intraspecific

genetic diversity, or cross-product interactions of intraspecific genetic diversity with environmental variables[68]) and Total Biomass on the one hand (model A) and Biomass Stability on the other hand (model B). To consider the influence of per capita biomass on total biomass variables, we hypothesized that intraspecific genetic diversity would also indirectly promote Total Biomass (respectively, Biomass Stability), through a pathway involving per capita Biomass (respectively, per capita Biomass Stability). We further hypothesized that the environmental characteristics of stations (upstream-downstream gradient UDG, eutrophic gradient EG and the corresponding cross-product interaction UDGxEG) would affect per capita and total biomass variables both directly (for instance through higher intraspecific competition in harsh conditions) and indirectly, through pathways involving intraspecific genetic diversity (promoted for instance by higher proximity to glacial refuges[25]) as well as bottleneck (triggered for instance by pollutants). All variables were standardized to z-scores before using weighted path analyses[22] with MLM maximum likelihood estimation, Satorra-Bentler scaled test statistic and station sampling weights. We simplified each model by removing non-significant paths one at a time, provided that cross-products were always associated with their additive terms[69] and that removal led to an increase in the relative fit of the model (i.e., a decrease in BIC score[70]). The validity of final models was assessed according to their absolute fit (standardized root mean square residual (SRMR) < 0.09 and Robust Comparative Fit Index (CFI) > 0.96)[71]. Additionally, we verified that there was no significant discrepancy between the sample and the fitted covariance matrices[71] (p-value associated with the model $\chi^2$ statistic > 0.05). All path analyses were run using the R-package lavaan[72]. Predicted values of endogenous variables (Fig. 3) were obtained from linear regressions with each beta coefficient fixed to the value of the corresponding direct link within the final model (Table 1). Error bands about predicted values (colored envelops in Fig. 3) stand for the standard deviation SD of 10,000 estimates, each obtained for a given predictor value through the random sampling of beta coefficients within their estimated distribution (mean and SD parameters being given by the final model).

## Variance partitioning

For each total biomass variable, we computed (a) the amount of variance ($R^2$) explained by each model. To assess the relative contribution of per capita biomass, intraspecific genetic diversity and environment to the variance in total biomass variables, we computing $R^2$ from further simplified models with (b) all variables related to per capita biomass being discarded (amount of variance explained by both environment and intraspecific genetic diversity), then (c) with all variables related to intraspecific genetic diversity (intraspecific genetic diversity and associated cross-products) being discarded (amount of variance explained by environment only). The relative contributions of per capita biomass and intraspecific genetic diversity to the variance in total biomass variables were respectively obtained by subtracting $R^2$ of (b) from $R^2$ of (a) and by subtracting $R^2$ of (c) from $R^2$ of (b).

## Overall expected change in biomass stability

To predict the overall expected change in biomass stability in our system given a 6 to 15% decline in intraspecific genetic diversity (IGD), as estimated by several authors in wild organisms[23,24], we used the following approach. Predictions were realized over $k = 10,000$ iterations. For each iteration $k$, we first computed an eroded IGD predictor (eIGD) as:

$$eIGD_i = IGD_i - E_i \times \max(IGD) \tag{1}$$

with $IGD_i$ the observed IGD level at station i and $E_i$ an erosion factor randomly sampled from a uniform distribution ranging from 0.06 to 0.15. For each iteration $k$, we then computed the predicted Biomass Stability BSTA from IGD and the predicted eroded Biomass Stability

eBSTA from eIGD, using linear models and Eutrophication gradient EG as a covariate (considering final causal model B; see Fig. 1 and Table 1) as follows:

$$BSTA_{ki} = \beta_{ki}^{IGDmaxBSTA} \times IGD_{ki} + \beta_{ki}^{EGmaxBSTA} \times EG_{ki} \qquad (2)$$

$$eBSTA_{ki} = \beta_{ki}^{IGDmaxBSTA} \times eIGD_{ki} + \beta_{ki}^{EGmaxBSTA} \times EG_{ki} \qquad (3)$$

with $BSTA_{ki}$, $eBSTA_{ki}$, $EG_{ki}$, $IGD_{ki}$ and $eIGD_{ki}$ standing for the expected biomass stability, the expected eroded biomass stability, the observed eutrophic level, the observed IGD level and the previously computed eroded IGD level at station $i$, respectively; with $\beta_{ki}^{IGDmaxBSTA}$ the effect of IGD on BSTA at station $i$ as sampled from a normal distribution of mean $\mu = 0.407$ and $\sigma = 0.147$ (corresponding to the SD of the path coefficient linking IGD to BSTA; Table 1); with $\beta_{ki}^{EGmaxBSTA}$ the effect of EG on BSTA at station $i$ as sampled from a normal distribution of mean $\mu = -0.220$ and $\sigma = 0.112$ (corresponding to the SD of the path coefficient linking EG to BSTA; Table 1). For each iteration $k$, we finally collected the raw mean difference $D_k$ between the predicted eroded Biomass Stability eBSTA and the predicted Biomass Stability BSTA as:

$$D_k = \overline{eBSTA_{ki}} - \overline{BSTA_{ki}} \qquad (4)$$

The predicted overall expected change in biomass stability given a 6 to 15% decline in IGD was finally computed as $\overline{D_k}$, with 95% quantiles as confidence interval.

### Reporting summary

Further information on research design is available in the Nature Portfolio Reporting Summary linked to this article.

## Data availability

The raw sequence data reported in this paper have been deposited in the Genome Sequence Archive (Genomics, Proteomics & Bioinformatics 2021)[73] in National Genomics Data Center (Nucleic Acids Res 2022)[74], China National Center for Bioinformation / Beijing Institute of Genomics, Chinese Academy of Sciences under the BioProject accession number PRJCA017984. All datasets supporting the results are deposited on Figshare (https://doi.org/10.6084/m9.figshare.13095380.v9)[38]. The ASPE database is available from the Zenodo repository (https://doi.org/10.5281/zenodo.7099129)[75]. The SIE database is available at https://adour-garonne.eaufrance.fr/. Source data are provided with this paper.

## Code availability

Code used to perform path analyses and produce figures[38]: Figshare https://doi.org/10.6084/m9.figshare.13095380.v9.

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

## Acknowledgements
This work was financially supported by the Office Français pour la Biodiversité (OFB) and the Agence Nationale pour la Recherche (SB). We warmly thank all the colleagues and students who helped with field sampling and data analyses.

## Author contributions
J.G.P., N.P. and S.B. formulated the main idea. S.B., G.L., C.V. and N.P. collected biological samples. J.G.P. collected environmental data and performed genetic/genomic analyses. M.C. and N.P. collected and prepared biomass data. J.G.P. performed data-analyses. J.G.P, S.B. and A.R. interpreted the data. J.G.P. and S.B. wrote the first draft and all authors provided feedback on subsequent versions of the manuscript.

## Competing interests
The authors declare no competing interests.
