## [Peer Review File · Nature Communications]

Genetic erosion reduces biomass temporal stability in wild fish populationsEditorial Note: This manuscript has been previously reviewed at another journal that is not operating a transparent peer review scheme. This document only contains reviewer comments and rebuttal letters for versions considered at Nature Communications.

RESPONSE TO REVIEWERS

Reviewer #1 (Remarks to the Author):

Thank you for the huge efforts you put in taking into account my comments and those of the two other reviewers. I greatly appreciated the added data (Fig 2), the clear explanations regarding the methods in the rebuttal and overall how the authors downplay their claim. I also think that modifications made improved the manuscript quality.

> We thank you for this positive feedback.

Reviewer #3 (Remarks to the Author):

The authors have responded extensively to my comments. I feel it is likely not constructive for me to write a full 'response to all responses', so I will focus on just 2 points:

1- I agree with the authors that there is a level of subjectivity in what is a "large" river system or what is "extensive sampling" and that very interesting studies can be conducted even on single populations. However, I do not agree that a comparison with Darwin's finches or Soay sheep is warranted here: those systems have extensive individual-level data over many decades. This study has population-level biomass data across a mean survey duration of 21 years, plus genetic data from 2 years. Such a level of information is common for many commercially important fish species (but with much higher quality genetic data). What this study does offer is such data in non-commercially important species, which is much rarer. But in terms of the extensiveness of the river system, the Garonne is ranked as the 47th longest river in Europe, and 30th by area and the Dordogne is lower https://en.wikipedia.org/wiki/List_of_rivers_of_Europe#Rivers_of_Europe_by_length . So while I agree that they are "not that small", it is also true that they are not that big. But this is not a "deal-breaker" by any means.

> We agree that what is actually new in this dataset is to obtain such data (long-term demographic data coupled to genetic and genomic data) in non-commercial species, i.e., species that are not exploited by humans but that are yet important from a conservation perspective. We now included this information in L. 61 to make clear that our study focuses on non-commercial fish.

2- The deal-breaker for me remains the genetic data on which the entire study is based. The number of loci per species (13-17 loci per species) is at a level that was commonly used in the 1990s and early 2000s in population genetic studies. Such low numbers of loci are unheard of in the majority of population genetic studies from recent decades. I highlighted the known limitations with the bottleneck detection methodologies used in my original review, and the authors responded by citing a simulation study they had conducted in 2013, published in Molecular Ecology. But I have a number of issues with this simulation study: for example, the conclusion that 'M-ratio test is reliable' is based on testing of empirical, and not the simulated data:

From the 2013 paper:

"We detected significant population bottlenecks for all

species in the two rivers when we analysed the empirical data. Because two of the three methods (MSVAR and M-ratio methods) were concordant in highlighting significant bottlenecks, we could reasonably assume that these populations had actually experienced demographic declines."

> *"The conclusion that 'M-ratio test is reliable' is based on testing of empirical, and not the simulated data"* (about the reference to Paz-Vinas et al.).

This is not completely true, the conclusions from Paz-Vinas et al. are also based on simulations, as quoted below:

- *"This analysis confirmed that the M-ratio method detected false signals of bottlenecks, but only for symmetric gene flow, and under some specific combinations of N and m".*
- *"False signals of expansion were detected [with MSVAR] under symmetric gene flow and with high migration rate ($m = 0.1$), but also under asymmetric gene flow ($a = 7.5$ or 50) and low to medium migration rates ($m = 0.01$ or 0.053)"*

In other words, Paz-Vinas et al. demonstrated using simulations that, in rivers, microsatellite-based bottleneck detection methodologies are rather robust to the false detection of bottlenecks and actually tend to detect false signals of demographic expansion. We now expand further the discussion to explicitly explain that false demographic changes can be inferred from microsatellite data, that in most cases it turns to be false signal of contractions, but that in rivers it is mostly false signals of expansion. We also discuss the potential implications of these potential biases on our finding. See L.183-192: *"This finding must be considered with caution, since recent bottleneck probabilities were estimated using a modest number of microsatellite markers, which can generate bias in demographic inferences³²⁻³⁵. Specifically, the limited number of loci sampled in the genome³² and the departure from the assumed mutation model^{33,34} or from mutation-drift equilibrium³⁵ generally increase the probability of inferring false signals of bottlenecks, i.e., of detecting a population decline in a truly stable population. This type of bias could have inflated the relationship we found between bottleneck probability and patterns of intraspecific genetic diversity. However, Paz-Vinas et al.³⁶ demonstrated using simulations that, in river systems, demographic inferences based on microsatellite markers are rather robust to this bias, and actually more likely to detect false signals of expansion, i.e., detect a population expansion in a truly stable population, than to detect false signal of bottlenecks."*

The fact that it was consistent with a second test that has also been criticised (e.g. here, where the M-ratio test is further criticised (Putman, A. I., & Carbone, I. (2014). Challenges in analysis and interpretation of microsatellite data for population genetic studies. *Ecology and Evolution*, 4(22), 4399–4428. <https://doi.org/10.1002/ece3.1305>) is not very convincing: Two potentially flawed tests giving the same result does not validate the reliability of one of these tests.

> *"The fact that [the Mratio] was consistent with a second test that has also been criticised [...] is not very convincing"*

If we understand this comment properly, Reviewer #3 here refers to the use of MSVAR in addition to the M-ratio in Paz-Vinas et al. 2013. We agree that the interpretation of two methods yielding consistent outputs may not be very convincing, when one of the two methods is known to be flawed. However, as explained above, the M-ratio was found robust to spurious demographic inferences when applied to rivers. Furthermore, we did not use

MSVAR but MIGRAINE and VAREFF, two methods that have been shown to perform better than MSVAR, especially in spatially structured populations (see original publications for details). By combining these methods, we thus hope to have limited any potential biases in demographic inferences (L.192-194: *“to somehow limit the potential biases associated with the use of microsatellite markers, past bottleneck inferences were based on three independent methods that yielded congruent estimates (Supplementary Note 2)”*). We would have been less confident with only 2 out of 3 congruent tests, or worse, with no congruent test at all. This congruence suggests that a pattern exists in the data, that can be partly captured by different methodologies based on very different assumptions, which is quite encouraging. Nevertheless, although we are confident in our inferences, we stay cautious in our discussion about the links between intraspecific genetic diversity and bottleneck probability (L195-202: *“Although the links between past-demographic events and contemporary patterns of intraspecific genetic diversity would merit further confirmation based on genomic data³², we believe that it is reasonable to consider the low levels of intraspecific genetic diversity observed in some populations to stem -at least in part- from recent, possibly human-induced, demographic contractions, for instance triggered by water eutrophication (Fig. 2 and 3A). Although alternative and non-exclusive historical processes (e.g., post glacial colonization events) are also likely to explain observed spatial patterns of intraspecific genetic diversity^{25,37}, this suggests that contemporary evolutionary processes can modulate ecological dynamics in natural settings. “*)

In the revised version of this current study, the authors further added in additional tests with yet new methods, and show they give relatively consistent results, but the fact remains that the low number of points sampled in the genome (loci) will limit reliability (<https://pubmed.ncbi.nlm.nih.gov/23967455/>). The fact that null alleles were detected at some loci, and the high drop-out rate during filtering of RAD loci adds further suspicions of DNA quality issues that could further bias results.

> *“[...] but the fact remains that [...] null alleles were detected at some loci, and the high drop-out rate during filtering of RAD loci adds further suspicions of DNA quality issues that could further bias results.”*

Reviewer #3 concluded that because we dropped out a high number of loci, then our data are of a poor quality. We do have good DNA quality data. We just employed strict and conservative filtering approaches to get a final dataset as irreproachable as possible. There are not many research groups that perform and describe filtering steps as we do in all our papers. Our protocols are transparently detailed in Supplementary Note 4, so that readers can know each step, its rationale, and the quality of final markers. We insist on the facts that (i) such filtering protocols are not systematically reported and (ii) we paid particular attention to the production of reliable pool-seq SNP allelic frequencies, with pool sample duplicates of all populations in two of the three species, which is actually rarely -if ever- done.

I am not, therefore, convinced that the results can be trusted, nor should they need to be trusted in this age of genomics.

1. *“I am not, therefore, convinced that the results can be trusted, nor should they need to be trusted in this age of genomics.”*

We disagree with this statement. Scientists handling microsatellite markers benefit from decades of experience (and simulation studies) so that pros and cons are exceptionally well known (as Reviewer #3 so aptly recalled), which is actually not yet the case when handling SNP markers. SNPs also have their own limitations (e.g., ascertainment bias; Nielsen and Signorovitch 2003; 10.1016/S0040-5809(03)00005-4) which are equally worrying, if not more, since (user-friendly) methods to circumvent them are yet to be developed (Dokan et al. 2021; 10.1093/g3journal/jkab128). Getting access to genomic data is extremely important, but -to quote Schlötterer-: "*recently developed high-throughput methods might not be unconditionally superior to more traditional approaches*" (Schlötterer 2004; 10.1038/nrg1249). We strongly believe in "*the age of scientific rigor*", where everything is done to obtain results that are both reliable and transparent, given the data and methods available, rather than where everything is done blindly to respond to the diktat of novelty. Microsatellite markers have crucial advantages for inferring neutral processes (high mutation rate, high allelic diversity) that come with limits (low number of markers in the genome). SNPs also have their own advantages (numerous loci across the genome with a very well-understood mutation model) and limits (bi-allelic markers, complex production process and associated ascertainment biases for instance). We remind Reviewer #3 that the central metric in our study (intraspecific genetic diversity) was estimated combining both microsatellites and SNP genomic data. We believe this makes our conclusions regarding the link between genetic diversity and biomass stability (i.e., the main finding of our MS) robust and trustworthy.

REVIEWERS' COMMENTS

Reviewer #1 (Remarks to the Author):

I was asked to review the manuscript and dialogue between the Authors and Reviewer 3 of this manuscript, and provide my recommendations on the difference of opinion. I have subsequently reviewed both the most recent exchange between authors and reviewers and the manuscript itself.

The primary issues here concern both the data themselves and the analytical paradigm (m-ratio/MSVAR tests, specifically).

Reviewer 3 contends, not incorrectly, that genome-scale SNP datasets consisting of thousands of loci are superior to small numbers of microsatellites.

The authors contend, not incorrectly, that microsatellite data have long been useful in identifying population bottlenecks, and that they have taken a conservative approach that recognizing known shortcomings of microsatellites.

First, I'll unpack the issue of microsatellites vs. genome-scale SNP data. Numerous empirical studies have demonstrated that SNP data are more sensitive (e.g. Zimmerman et al., 2020) and more cost effective (e.g. Puckett, 2017, but note that cost effective does not mean less costly). In the case of the former proposition, SNPs have indeed been shown to be able to elucidate a finer-grained population genomic perspective when compared to microsatellites. However these studies have not shown that microsatellites are unable to elucidate population genetic-level information or that the results derived from them are spurious. In fact, before the modern population genomics era, microsatellites were long relied upon to draw inferences related to population genetic patterns and processes. To suggest otherwise is an abrogation of our population genetic history.

With respect to the second proposition, indeed population genomic data in the form of SNPs are more cost effective in terms of the information (both at putatively neutral sites, and sites likely under selection, greatly expanding on insights that can be gleaned), but that's not to say that, on balance, it is less expensive. Thus, setting genome-scale data as a bar for entry to publishing is an affront to just, equitable, and inclusive science, both with respect to developing nations and early career scientists.

Given the above, I reject the notion that microsatellite data are inherently flawed because they're not genome-scale. Would genome-scale be better? Sure. Is that always feasible? Absolutely not.

With respect to the analytical issues, Reviewer 3 astutely points out the "potential" issues with the m-ratio test, as well as "potential" issues with MSVAR. The authors conduct a third test that yields, again, consistent results. And while the concerns over these results are valid, at no point to the authors try and obfuscate or obscure these issues. The authors have couched the matter appropriately and are transparent in the "potential" implications of these "potential" issues. Their approach is conservative and measured, cognizant of "potential" flaws and pitfalls.

Given the above, I recommend the article be accepted.

REVIEWERS' COMMENTS

Reviewer #1 (Remarks to the Author):

I was asked to review the manuscript and dialogue between the Authors and Reviewer 3 of this manuscript, and provide my recommendations on the difference of opinion. I have subsequently reviewed both the most recent exchange between authors and reviewers and the manuscript itself.

The primary issues here concern both the data themselves and the analytical paradigm (m-ratio/MSVAR tests, specifically).

Reviewer 3 contends, not incorrectly, that genome-scale SNP datasets consisting of thousands of loci are superior to small numbers of microsatellites.

The authors contend, not incorrectly, that microsatellite data have long been useful in identifying population bottlenecks, and that they have taken a conservative approach that recognizing known shortcomings of microsatellites.

First, I'll unpack the issue of microsatellites vs. genome-scale SNP data. Numerous empirical studies have demonstrated that SNP data are more sensitive (e.g. Zimmerman et al., 2020) and more cost effective (e.g. Puckett, 2017, but note that cost effective does not mean less costly). In the case of the former proposition, SNPs have indeed been shown to be able to elucidate a finer-grained population genomic perspective when compared to microsatellites. However these studies have not shown that microsatellites are unable to elucidate population genetic-level information or that the results derived from them are spurious. In fact, before the modern population genomics era, microsatellites were long relied upon to draw inferences related to population genetic patterns and processes. To suggest otherwise is an abrogation of our population genetic history.

With respect to the second proposition, indeed population genomic data in the form of SNPs are more cost effective in terms of the information (both at putatively neutral sites, and sites likely under selection, greatly expanding on insights that can be gleaned), but that's not to say that, on balance, it is less expensive. Thus, setting genome-scale data as a bar for entry to publishing is an affront to just, equitable, and inclusive science, both with respect to developing nations and early career scientists. Given the above, I reject the notion that microsatellite data are inherently flawed because they're not genome-scale. Would genome-scale be better? Sure. Is that always feasible? Absolutely not.

With respect to the analytical issues, Reviewer 3 astutely points out the "potential" issues with the m-ratio test, as well as "potential" issues with MSVAR. The authors conduct a third test that yields, again, consistent results. And while the concerns over these results are valid, at no point to the authors try and obfuscate or obscure these issues. The authors have couched the matter appropriately and are transparent in the "potential" implications of these "potential" issues. Their approach is conservative and measured, cognizant of "potential" flaws and pitfalls.

Given the above, I recommend the article be accepted.

> We warmly thank you for your effort and your understanding. We believe that our difference of opinion with Reviewer 3 reflects an important issue in molecular ecology today: can reliable knowledge be produced from non-genomic data? We think so, and you elegantly but firmly asserted the same, which we greatly appreciate: we hope that this decision concerning our manuscript can weigh in the Microsatellite/SNP debate in the future.